# The Value of Hemoglobin Glycation Index–Diabetes Mellitus System in Evaluating and Predicting Incident Stroke in the Chinese Population

**DOI:** 10.3390/jcm11195814

**Published:** 2022-09-30

**Authors:** Pengbo Wang, Qiyu Li, Xiaofan Guo, Ying Zhou, Zhao Li, Hongmei Yang, Shasha Yu, Yingxian Sun, Xingang Zhang

**Affiliations:** Department of Cardiology, The First Hospital of China Medical University, 155 Nanjing North Street, Heping District, Shenyang 110001, China

**Keywords:** hemoglobin glycation index, diabetes mellitus, stroke, follow-up study

## Abstract

We aimed to clarify the effect of the hemoglobin glycation index (HGI)–diabetes mellitus (DM) system in evaluating the risk of incident stroke. We followed up on 2934 subjects in rural regions of Northeast China, established Cox proportional hazards models to evaluate the effects of the HGI–DM system in describing stroke risk, and further conducted a discrimination analysis to confirm the improvement in HGI based on the traditional stroke risk model. After a median of 4.23 years of follow-up, 79 subjects developed stroke or related death. DM-high HGI condition significantly elevated the risk of incident stroke (hazard ratio (HR): 2.655, 95% confidence interval (CI): 1.251–5.636). In addition, higher HGI levels elevated the risk of stroke, even if the patients did not have DM (HR: 1.701, 95% CI: 1.136–2.792), but DM failed to bring an extra risk of incident stroke to patients with lower HGI levels (HR: 1.138, 95% CI: 0.337–3.847). The discrimination analysis indicated that the integrated discrimination index (IDI) of the HGI model was 0.012 (95% CI: 0.007–0.015) and that the net reclassification index (NRI) was 0.036 (95% CI: 0.0198–0.0522). These results indicated HGI was associated with the onset of stroke, and high HGI indicated an aggravated trend in glycemic status and increased risk of incident stroke. The HGI–DM system enabled us to identify the different glucose statuses of patients, to conduct suitable treatment strategies, as well as to improve the predictability of incident stroke based on the traditional model.

## 1. Introduction

Stroke is an acute cerebrovascular disease that is characterized by high morbidity, mortality, and disability [1,2,3,4]. In recent decades, the incidence of stroke has increased by 70.0%, stroke-related deaths have increased by 43.0% worldwide, and more than 6 million patients die from stroke each year [5]. Recently, stroke has become the leading cause of death and disability and was recognized as the disease with the highest disability-adjusted life-years lost in China [6], especially in Northeast China’s rural regions, where stroke is the leading disease burden [2,7,8]. Furthermore, with the gradual increase in the prevalence and decreased onset age of stroke, risk factors have become more multifaceted and complicated, and the traditional risk model is no longer suitable for evaluating the risk of stroke [9]. Thus, it is necessary to conduct a novel systematic stratification and comprehensive risk assessment to identify the potential populations with a high risk of stroke so that we can provide different and individualized preventive measures for different risk subgroups and thus reduce the disease burden.

Diabetes mellitus (DM) is recognized as a classical risk factor for stroke. Some studies have observed that patients with DM have a 2–4 times higher stroke risk than non-DM patients [10,11], and even impaired serum glucose status can increase the risk of stroke [12]. However, the current diagnostic criteria of DM are mainly based on fasting blood glucose (FPG) levels, but the use of DM alone as a method of serum glucose status evaluation has some limitations [13,14], as it can only reflect the serum glucose status of individuals in the short term and has a poor ability to predict long-term adverse outcomes [15]. Glycated hemoglobin (HbA1c) is a product of hemoglobin glycosylation that can reflect long-term serum glucose alterations in patients, has recently received increased attention, and has already been used in guidelines for serum glucose management [16]. However, HbA1c also has some limitations, such as individual differences, in which HbA1c can be easily affected by other diseases and physical status alterations [17,18]. Hence, HbA1c is not effective for evaluating serum glucose independently.

On this basis, the concept of HGI was proposed, which described the gap between the predicted HbA1c and actual measured HbA1c [19], and this parameter can effectively reduce the individual differences in HbA1c and indirectly reflect the trend of glycosylation when evaluating the glycemic status of patients [20]. Several studies have found a potential relationship between HGI levels and incident risk or prognosis of stroke in diabetic patients [21,22], suggesting that HGI might be able to optimize serum glucose evaluation for populations with high stroke risk. DM represents the patient’s glycemic status, while HGI represents the patient’s glycosylation tendency, which can further evaluate the patient’s prognosis and predict disease progression based on the present glycemic status, so we hypothesized that the DM-HGI system could also play a positive role in the disease risk evaluation process. However, no study has yet combined the DM and HGI together to evaluate patients’ glycemic status and the process or prognosis of other chronic diseases.

The present study was based on a prospective cohort of a rural population in Northeast China and combined HGI and DM for the first time to evaluate the effect of the HGI–DM system in predicting and identifying potential patients with a high risk of incident stroke and provided evidence for the theory that patients with different glycemic statuses should receive different interventions and prevention strategies. The results of the present study might provide support for the comprehensive evaluation of serum glycemic status in combination with HGI levels as a novel and effective method to identify and classify the population with stroke risk at an early stage. This can enable practitioners to carry out suitable and timely management and prevention strategies for different populations according to their different risk characteristics.

## 2. Methods

### 2.1. Study Population

The present study is based on The Northeast China Rural Cardiovascular Health Study (NCRCHS) to further reveal the correlation between HGI and incident stroke. We conducted the baseline study from July 2012 to August 2013, and we conducted the follow-up study on these participants in 2017; the protocol details were described in previous research [23,24], and they are summarized in Figure 1. Additionally, the inclusion and exclusion criteria were as follows: inclusion criteria: (i) aged ≥ 35 years; (ii) permanent residents of the regions; (iii) can sign an informed consent form on their own. Exclusion criteria: (i) pregnancy; (ii) cancer; (iii) mental disorders; (iv) expected survival time of less than 5 years. At baseline, we recruited a total of 11,956 natural residents (age ≥ 35 years) from rural regions in three counties in Northeast China. In 2017, we invited all these participants to attend the follow-up study, and 10,700 participants consented and were qualified for our follow-up study. In the present study, we excluded 6858 participants who lacked related information or missed the follow-up and further excluded 908 participants who were suffering or had suffered from a stroke at baseline; eventually, we obtained a target population of 2934 for the present study.

### 2.2. Ethics Approval and Informed Consent

The Ethics Committee of China Medical University approved our project and the following research based on the present project in 2012 (Shenyang, China, ethical approval project identification code: AF-SOP-07-1, 0-01), and all procedures met Declaration of Helsinki Standards. Furthermore, every participant received and signed a paper version informed consent after clarifying relevant information in terms of the study objectives, benefits, medical procedures, confidentiality, agreement on personal information, and agreement on the publication of the present study and subsequent research.

### 2.3. Data Collection and Lifestyle Risk Factors

The data collection for this study was based on outpatient face-to-face interviews and paper version standard questionnaires conducted by a team of cardiologists. The study population received and signed a paper version informed consent form after clarifying relevant information and the study objectives, benefits, medical procedures, and confidentiality agreement on personal information. Information such as age and gender were obtained from a standard questionnaire in a face-to-face interview. All participants were asked whether they had a history of cardiovascular diseases (CVD) and were currently smoking or drinking.

### 2.4. Anthropometrics, Biochemical Parameters, and Blood Pressure Measurements

All participants were told to fast for at least 12 h in advance and collected blood samples the next morning. The blood samples were added to vacutainer tubes containing anticoagulants and obtained plasma by centrifuging. FPG, triglyceride (TG), plasma total cholesterol (TC), low-density lipoprotein cholesterol (LDL-C), high-density lipoprotein cholesterol (HDL-C), and HbA1c were obtained by Enzymatic analysis on an Olympus AU640 automated analyzer (Olympus, Kobe, Japan). We calibrated all laboratory equipment and conducted repeat measurements on these samples three times using the blind method to obtain the average of the results for subsequent analysis. Measurements of height and weight required participants to wear light clothes, remove shoes, and maintain a standing position. The measurement results were accurate to 0.1 kg and 0.1 cm, respectively. The measurement of blood pressure was according to the American Heart Association protocol [25]. Subjects were asked to rest for at least 10 min in a quiet room in a seated position, then use an automated electronic sphygmomanometer (HEM-741C; Omron, Tokyo, Japan) to measure blood pressure on the bare upper arm three times with a 5 min interval between measurements. We took the average of three blood pressure measurements and used them for all subsequent analyses.

### 2.5. Definition

According to 7th Joint National Committee (JNC7), we defined hypertension as the following conditions: systolic blood pressure (SBP) ≥ 140 mmHg and (or) diastolic blood pressure (DBP) ≥ 90 mmHg, or reported use of a medication for hypertension [26]. Diabetes was defined as a fasting plasma glucose (FPG) ≥ 7.0 mmol/L or a previous diagnosis of diabetes [27]. The body mass index (BMI) was calculated according to the following formula: BMI = Weight(kg)/Height^2^(m^2^). The estimated glomerular filtration rate (eGFR) was calculated using the Chronic Kidney Disease Epidemiology Collaboration equation [28]. The linear relationship between HbA1c and FPG was estimated from the linear regression analysis of the study subjects’ data and the predicted HbA1c = 0.433 × FPG + 2.809. Additionally, HGI was calculated as the difference between the actual and theoretical level of HbA1c; the detailed formula was the following: HGI = measured HbA1c-predicted HbA1c [20,29]. In addition, we divided HGI into two levels, HGI < 0 and HGI ≥ 0.

### 2.6. Judgment and Definition of Clinical Outcomes

The median follow-up period was 4.23 years. In the present study, the outcome event was defined as a new onset of stroke during the follow-up period. For all participants reporting possible diagnoses, all available clinical-information-related materials were collected and independently adjudicated by the Endpoint Assessment Committee. The materials included medical records and imaging examinations. According to the WHO Multinational Monitoring of Trends and Determinants of Cardiovascular Disease (MONICA) criteria [30], stroke was defined as signs of rapidly developing focal or global disturbance of cerebral function that lasts for longer than 24 h (unless the patient undergoes surgery or dies). In addition, transient ischemic attacks and chronic cerebrovascular disease were excluded.

### 2.7. Statistical Analysis

Overall, the data were normally distributed; thus, the continuous variables were described in the format of the mean value (M) and standard deviation (SD), and categorical variables were described by frequency and percentage. As appropriate, differences between categories were evaluated using the one-way analysis of variance (ANOVA) test (for continuous variables) or the χ^2^ test (for categorical variables). Meanwhile, we established the Cox proportional hazards models to calculate hazard ratios (HRs) and 95% confidence intervals (CIs) to evaluate the relationship between the risk of incident stroke and different glucose statuses. We drew Kaplan–Meier curves to exhibit the tendency of cumulative hazard, which was stratified by the DM-HGI system, and conducted a log-rank test to compare the differences. We drew restricted cubic spline (RCS) curves and conducted a linear test to evaluate the correlation between the HGI level and the risk of incident stroke, further describing the effect of HGI in evaluating the risk of stroke according to the HR trends in the curve. To further evaluate the potential of HGI to improve the identification of incident stroke with traditional clinical risk factors, we calculated the integrated discrimination index (IDI) and net reclassification index (NRI). The Kaplan–Meier curves and relevant analysis were performed using “survminer packages” and “survival packages”, the RCS analysis was conducted using “rms packages”, and re-stratification analysis was performed using “survIDINRI packages” and “survival packages” of R software (version 4.0.5, http://www.R-project.org, accessed on 23 April 2022. The R Foundation). Statistical analyses were performed using SPSS software version 22.0 (IBM Corp., Armonk, NY, USA); *p* < 0.05 under the two-tailed condition was considered statistically significant.

### 2.8. Independent Data Access and Analysis

The corresponding author Xingang Zhang and co-author Yingxian Sun have full access to all the data in the study and take responsibility for their integrity and the data analysis. All co-authors read and approved the final manuscript. Moreover, all co-authors and participants gave their consent for publication of this article.

### 2.9. Materials and Data Availability

We can provide the raw data of the present study after evaluation and permission from the subject principals. For matters on the availability of raw data, please contact Professor Xingang Zhang (zhangxingang80@aliyun.com) and Professor Yingxian Sun (yxsun@cmu.edu.cn).

## 3. Results

### 3.1. Baseline Characteristics of the Study

Table 1 summarizes the characteristics of participants at baseline. The current analysis enrolled 2934 participants with a mean age of 55.5 ± 10.0 years, and 45.6% of participants were male. DM was present in 10.7%, and around 58.3% (n = 183) also had high HGI levels. In total, DM patients had higher BMI, blood pressure, FPG and HbA1c, TC, TG, and LDL; had lower eGFR and HDL; and were more likely to suffer from CVD or hypertension. However, we noticed that the population characteristics of the low-HGI group were close to those of the high-HGI population. Based on these significant differences, we recognized these risk factors as independent variables to build the multivariate Cox proportional hazards model to further clarify or eliminate the impact of these factors on the association between HGI and incident stroke.

### 3.2. The Incidence and Cumulative Hazard of Onset Stroke Based on the HGI–DM System

In the 4.23 years of the follow-up period, 79 participants (2.7%) developed stroke events (crude incidence rate: 6.30 incident stroke events per 1000 person-years, the median time of incident stroke: 1.92 years, 5–95% duration of incident stroke: 0.16–4.19 years). In detail, among those 79 participants, 21 subjects were in non-DM and low-HGI subgroups (21/1276), 3 subjects were in isolated DM subgroups (3/131), 47 were in isolated high-HGI subgroups (47/1344), and 8 subjects were in both DM and high-HGI subgroups (8/183). As shown in Figure 2, the highest incidence of stroke took place in the population with both DM and high HGI, with 10.37 incident strokes per 1000 person-years. Additionally, the population without either DM or high HGI had the lowest crude incidence (crude incident rate: 3.83 incident strokes per 1000 person-years). In addition, the crude incidence of stroke among the isolated high-HGI participants was slightly higher than the isolated DM participants (8.23 incident strokes and 5.35 incident strokes per 1000 person-years, respectively). In addition, we also drew Kaplan–Meier curves to exhibit the tendency of cumulative hazard and further compared different DM-HGI statuses using the log-rank test (Figure 3). We noticed that compared with the subjects who had neither DM nor high HGI, isolated DM patients did not present a higher stroke risk. However, two high-HGI subgroups both had a higher cumulative hazard of stroke than the reference group. Specifically, the isolated high-HGI subgroup still presented an elevated risk of stroke, although they did not suffer from DM. Additionally, patients with both DM and high HGI had the highest hazard of incident stroke.

### 3.3. High HGI Can Elevate the Risk of Incident Stroke and Promote Adverse Outcomes

Table 2 summarizes the association between the risk of incident stroke events and HGI levels, which was analyzed by Cox proportional hazard models. The risk-factor variables of the Cox proportional hazards model include age, gender, current smoking and drinking, hypertension, eGFR, TC, TG, HDL-C, LDL-C, and HGI or DM-HGI status. Every SD increment of HGI could bring an additional 27.9% risk of incident stroke (HR: 1.279, 95% CI: 1.063–1.539, *p* < 0.05). Meanwhile, the high-HGI group had a 1.78-fold greater risk of incident stroke than low-HGI group participants (HR: 1.780, 95% CI: 1.120–2.830, *p* < 0.05).

After further combining DM status and HGI levels, the participants were divided into four subgroups as follows: non-DM and low-HGI population (reference group), DM patients with low HGI, high HGI without DM, and concomitant DM and high HGI. As shown in Table 3 (also adjusted by the risk-factor variables mentioned before), among the low-HGI population, DM did not bring a significantly increased risk of incident stroke than the normal population. However, isolated high HGI levels increased the risk of incident stroke events by 70.1% compared to the participants in non-DM and low-HGI groups (HR: 1.701, 95% CI: 1.136–2.792, *p* < 0.05). Concomitant DM and high HGI increased the risk of incident stroke the most, with a 1.655-fold greater risk (HR: 2.655, 95% CI: 1.251–5.636, *p* < 0.05).

### 3.4. High HGI Trend Increases the Risk of Incident Stroke Regardless of DM

To further confirm the effect of the HGI–DM system in evaluating the risk of incident stroke, we divided participants into two subgroups, normal subjects (n = 2620) and DM subjects (n = 314), and further drew the RCS curve in each subgroup (Figure 3). We observed that the HGI indeed had a linear correlation with incident stroke risk in both subgroups, where the risk of incident stroke was increased gradually with elevated HGI levels; a low HGI level (HGI < 0) could reduce stroke risk; and high HGI (≥0), which represented a trend towards excessive glycosylation, could increase the risk of incident stroke. In addition, the RCS curve also revealed that high HGI (≥0) indeed had a weak but significant effect in increasing stroke risk, even in non-DM participants (Figure 4A), confirming the results we previously mentioned wherein isolated high HGI levels could still increase the risk of incident stroke by 70.1%, even if the subjects were non-DM patients.

### 3.5. HGI Can Improve the Predictive Effect of the Traditional Model for Stroke

Table 4 shows the discrimination analysis used to evaluate whether HGI could improve the risk stratification of incident stroke events. As we expected, we found a significant category-free IDI (0.012, 95% CI: 0.007–0.015, *p* < 0.001) and NRI (0.036, 95% CI: 0.0198–0.0522, *p* < 0.001) after adding HGI to the conventional model (including age, gender, current smoking and drinking, BMI, hypertension, DM, TG, TC, HDL, and LDL).

## 4. Discussion

We followed up on the rural population in Northeast China for 4.23 years, trying to reveal the epidemiological feature and alteration trends of CVD and related risk factors. The present study combined HGI and DM together for the first time to evaluate the glucose status based on DM alone and confirmed the complementary value of the HGI–DM system for the prediction of incident stroke in the general rural population. We revealed that HGI was an independent risk factor for incident stroke and not redundant based on DM. Our results have shown that the risk of incident stroke was not significantly increased in the population with low HGI, regardless of whether they were suffering from DM. On the contrary, in the subjects with high HGI, regardless of whether they had DM, the risk of incident stroke was still elevated significantly. These results revealed that excessive glycosylation might promote the incidence of stroke. In addition, we also found that HGI could significantly improve the predictability of incident stroke based on traditional models. Our study confirmed the capacity of the DM-HGI system in evaluating and predicting the risk of incident stroke and prompted us to believe that the glycosylation trend represented by the HGI might be a determinant of the risk of incident stroke, so it is effective and necessary to classify and manage high-risk individuals according to the HGI level based on DM and provide different management strategies. For patients with isolated DM (who have DM but without a high HGI level), we need to keep the current treatment and management strategies while actively monitoring their HGI levels and glycosylation trends. Meanwhile, we should track and follow up on patients with high HGI who might already have abnormal glycation, as the aggravated trend in glycosylation status can increase the risk of incident stroke. Tracking will help us promptly treat these patients with care and prevention. The DM-HGI system enabled us to conduct different treatment strategies for patients with different glycemic statuses, reducing incident stroke events at various levels.

DM was recognized as an important risk factor for stroke, so it was essential to comprehensively evaluate glycemic status and screen the high-risk crowds [31,32]. Current studies have mostly used FPG or random serum glucose levels to diagnose DM, but these can only reflect the short-term glucose level of individuals, so they have a poor predictive effect on long-term adverse events [15]. HbA1c is the product of hemoglobin glycosylation, which reflects the long-term serum glucose level. It is considered a gold standard for evaluating glycemic control status in DM patients and is used to predict the risk of DM and related complications [33]. However, some studies observed individual differences in the glycosylation process of hemoglobin, and some patients within the same FPG level could have different HbA1c degrees, with this difference potentially existing over time [17,18,34,35]. HGI reflects the degree of variation of HbA1c and represents the glycosylation trend of hemoglobin in the human body, which can quantify the degree of glycosylation and eliminate the individual differences of HbA1c [20]. Multiple studies have confirmed that HGI is suitable for evaluating the incident risk and outcomes prognosis of various chronic diseases [36,37], especially for DM-related complications, and could act as a predictive indicator of intensive hypoglycemic-strategy-related adverse reactions [29,38,39,40], which we also observed and obtained a similar conclusion for in Table 2. In addition, our studies observed that DM patients with high HGI (HGI > 0) had a significantly increased risk of stroke, the maximum incident risk could even reach 5.636-fold, and this conclusion was also supported by previous studies [22,35]. A randomized controlled trial of T2DM patients revealed that high HGI was associated with the risk of major adverse cardiovascular events [41]. A cohort study of DM and CVD found that a higher HGI level could bring a 1.94-fold increased risk of incident stroke [42]. In addition, a study indicated that DM patients with high HGI have a higher risk of stroke recurrence and a poor prognosis of stroke [22].

After combining DM status and HGI levels, two subgroups showed unexpected results, which might improve the management strategy in clinical treatment. Firstly, we observed that high HGI could elevate the risk of incident stroke, although the subjects did not have DM (HR: 1.701, 95% CI: 1.136–2.792, *p*: 0.036). HGI was the gap between the actual and theoretical level of HbA1c; it seemed to reflect the glucose metabolism reserve or islet function reserve in the human body [43]. The subjects who had normal or slightly elevated FPG levels should not have such high HGI levels, but they indeed had a higher HGI level, which suggested that excessive glycosylation was activated, and glucose metabolism disorder might already be active or further aggravated. In addition, our results indicated that HGI seemed more sensitive than FPG alone in describing the risk of incident stroke. A follow-up study showed that people with abnormal glucose levels had an increased risk of CVD when their HGI levels were increased, even in prediabetics, whose FPG was increased slightly and defined as impaired glucose status. In addition, they also observed that the risk of stroke was elevated from 2.31-fold to 3.40-fold compared to the reference, with a gradual increase in HGI levels [21]. This conclusion is similar to our study, which suggested that HGI could effectively reflect the health status of vascular metabolism, and we believe HGI might be better for evaluating the risk of CVD than DM (as exhibited in Table 3). Additionally, this hypothesis was confirmed in another subgroup of our study, in which we did not observe a significant difference in incident stroke risk among the low-HGI subjects regardless of whether they had DM, meaning that under the condition of low HGI, DM failed to bring an extra risk of incident stroke, and we also observed this tendency in the cumulative hazard or incidence of onset stroke (Figure 2 and Figure 3). This conclusion was supported by the Action to Control Cardiovascular Risk in Diabetes (ACCORD) trial. The ACCORD study included 10,251 T2DM patients who were given standard treatment or intensive hypoglycemic treatment, respectively. After 3.5 years of follow-up, there was a significant improvement in the prognosis of patients with low HGI and medium HGI, but there was no significant improvement in the prognosis of patients with high HGI, even if they were given intensive hypoglycemic strategy, and high HGI could bring higher total mortality or a greater risk for hypoglycemia [44]. Both our study and the ACCORD study revealed that lower HGI levels could improve the outcome events and benefit patients with a risk of stroke regardless of whether they were DM patients.

The present study has some strengths. First, our research conducted a population-based prospective cohort study with many participants in northeastern rural regions of China, which gave our conclusions and the causative association between HGI level and the risk of incident stroke a higher degree of evidence. Additionally, the present study used the DM-HGI system to first predict the risk of incident stroke; we found that this combination could divide the population into different glucose metabolic statuses to optimize the clinical treatment and enable us to conduct different management strategies. Lastly, our study, for the first time, incorporated HGI into the traditional risk model and conducted a re-stratification analysis to evaluate the improved ability to predict incident stroke, obtaining a positive effect, which is shown in Table 4. This enabled us to optimize the stroke risk prediction model and conduct relevant treatment in time. However, we also had some limitations. First, our variable information did not cover all aspects; for example, we did not collect drug consumption information and AF information for our regression model, which was defined as a crucial risk factor. Secondly, the present study excluded 6858 subjects who lacked relevant data, such as Hb1Ac and stroke status, in the follow-up period. Although we used the Kolmogorov–Smirnov test to confirm that the data were normally distributed and the present study population remained broadly representative, these exclusions might have led to selection bias, and we will perform a larger-scale cohort study to improve and confirm the present conclusion. Moreover, the outcome events of our study combined incident stroke and stroke-related death, weakening the predictive value of HGI for nonfatal stroke or relevant death events. Lastly, the HGI was calculated by a regression formula, which might not be suitable for other natural populations, so our conclusion still needs a multiple-center and large-scale study to confirm the results.

## 5. Conclusions

In summary, we confirmed the complementary value of the HGI–DM system for predicting incident stroke in the general rural population. Our results provided a novel insight into managing the high-risk crowds of stroke. The patients with low HGI suggested that total glucose metabolism was stable, and these patients should further keep this homeostasis. Patients with high HGI but without DM should receive more attention regarding their glucose metabolic status because they might already have abnormal glycation. Lastly, we found that HGI could effectively improve the identification of incident stroke based on the traditional risk model, in which HGI had potential applications in risk stratification and the identification of individuals at high risk for poor outcomes.

## Figures and Tables

**Figure 1 jcm-11-05814-f001:**
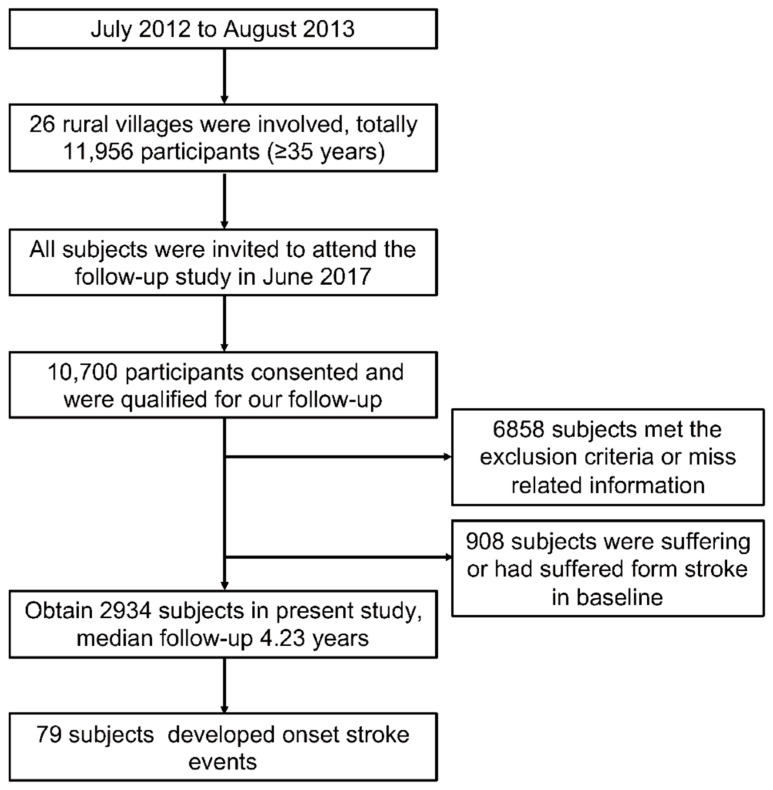
Flow chart of our study population’s selected protocol. We recruited a total of 11,956 natural residents (age ≥ 35 years) from rural regions in three counties in northeastern China. In 2017, 10,700 participants consented and were qualified for our follow-up study. In the present study, we excluded 6858 participants who lacked related information or missed the follow-up and further excluded 908 participants who were suffering or had suffered from stroke in baseline. Eventually, we obtained a target population of 2934 for the present study.

**Figure 2 jcm-11-05814-f002:**
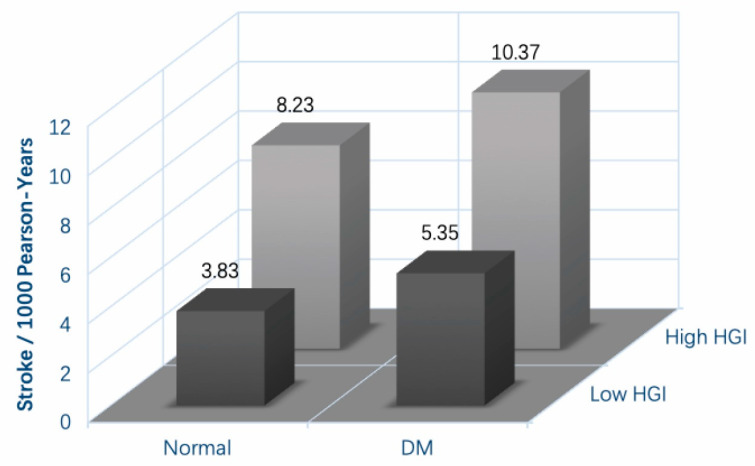
The incidence of onset stroke based on HGI–DM system. After 4.23 years of follow-up, based on HGI–DM system, the highest incidence of stroke took place in both DM and high-HGI subgroups. Moreover, the isolated high-HGI subjects had a slightly higher incidence than the isolated DM subjects. Abbreviations: HGI, glycated hemoglobin index; DM, diabetes mellitus.

**Figure 3 jcm-11-05814-f003:**
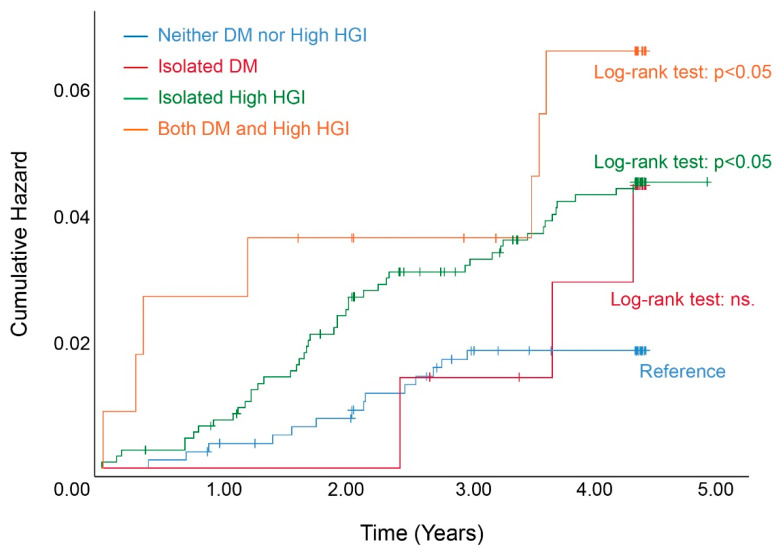
Kaplan–Meier curves for cumulative hazard of incident stroke events stratified by DM-HGI system. Kaplan–Meier curves exhibited the tendency of cumulative hazard, and we further compared different DM-HGI statuses using the log-rank test. Compared with the subjects who had neither DM nor high HGI (blue curve), isolated DM patients did not present a higher incident stroke risk (red curve). However, two high-HGI subgroups both had higher cumulative hazards of stroke than reference. Especially, the isolated high-HGI subgroup presented an elevated risk of stroke, although the subgroup population did not suffer from DM (green curve). Moreover, the patients with both DM and high HGI had the highest hazard of stroke (orange curve). Abbreviations: HGI, glycated hemoglobin index; DM, diabetes mellitus. Statistical significance was defined as *p* < 0.05 under two-tailed condition.

**Figure 4 jcm-11-05814-f004:**
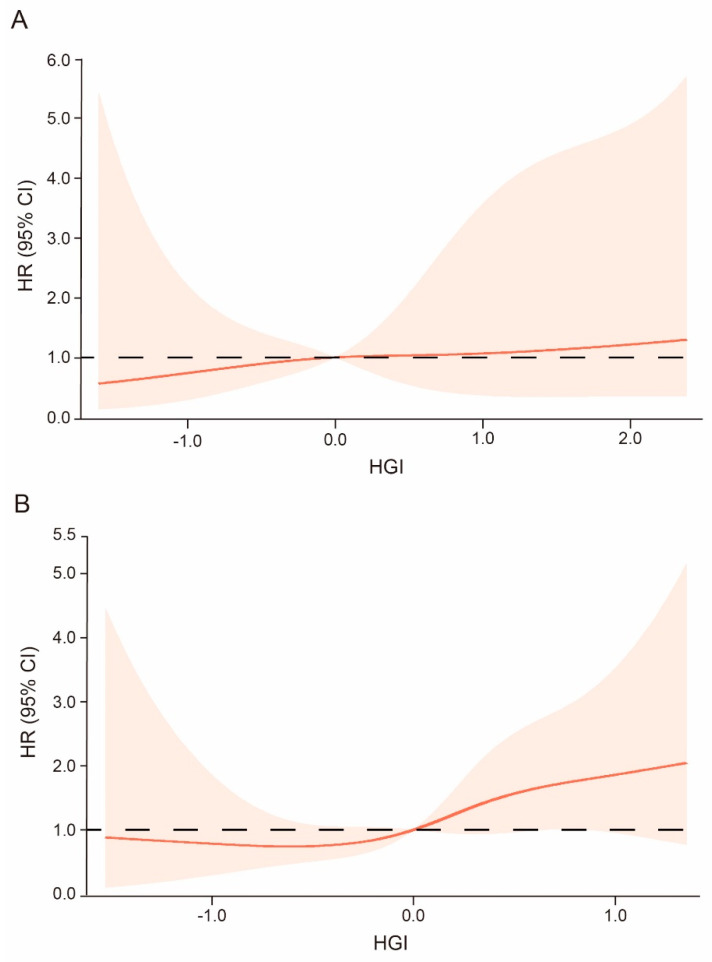
The RCS of HGI levels and incident stroke risk grouped by DM status. The HGI had a linear correlation with the risk of onset stroke, in which higher HGI could increase the risk of incident stroke. Meanwhile, even in the subjects without DM, high HGI still could increase the risk of incident stroke. (**A**). The RCS of HGI and risk of incident stroke in non-DM participants, n = 2620; (**B**). The RCS of HGI and risk of incident stroke in DM participants, n = 314. Abbreviations: HR, hazard ratio; CI, confidence interval; HGI, glycated hemoglobin index.

**Table 1 jcm-11-05814-t001:** Baseline characteristics of the study.

Variable	Normal Population		DM Patients		*p* Value
	Low HGI n = 1276	High HGI n = 1344	Low HGI n = 131	High HGI n = 183	
Age (years)	54.4 ± 10.3	56.0 ± 9.9	57.9 ± 9.3	58.1 ± 8.5	<0.001
Male	639 (50.1)	556 (41.4)	74 (56.5)	68 (37.2)	<0.001
Smoking	465 (36.4)	483 (35.9)	58 (44.3)	57 (31.1)	0.122
Drinking	322 (25.2)	202 (15.0)	43 (32.8)	23 (12.6)	<0.001
History of CVD	189 (14.8)	219 (16.3)	38 (29.0)	52 (28.4)	<0.001
Hypertension	516 (40.4)	544 (40.5)	85 (64.9)	118 (64.5)	<0.001
BMI (kg/m^2^)	24.06 ± 3.51	24.30 ± 3.84	25.77 ± 3.56	25.63 ± 3.68	<0.001
SBP (mmHg)	136.28 ± 21.39	135.77 ± 22.28	148.15 ± 24.34	147.81 ± 23.67	<0.001
DBP (mmHg)	81.33 ± 11.22	80.40 ± 11.74	84.01 ± 13.20	83.98 ± 11.90	<0.001
FPG (mmol/L)	5.61 ± 0.53	5.42 ± 0.52	9.46 ± 2.92	9.08 ± 2.65	<0.001
TC (mmol/L)	5.05 ± 1.02	5.12 ± 1.04	5.27 ± 1.20	5.45 ± 1.13	<0.001
TG (mmol/L)	1.74 ± 1.66	1.67 ± 1.44	2.73 ± 2.49	2.56 ± 2.32	<0.001
HDL-C (mmol/L)	1.32 ± 0.29	1.31 ± 0.29	1.26 ± 0.29	1.26 ± 0.26	0.004
LDL-C (mmol/L)	2.61 ± 0.62	2.71 ± 0.68	2.67 ± 0.64	2.85 ± 0.70	<0.001
eGFR (mL/min/1.73 m^2^)	89.35 ± 13.34	87.34 ± 12.57	86.18 ± 16.03	85.78 ± 13.12	<0.001
HbA1c (%)	4.73 ± 0.49	5.59 ± 0.38	6.11 ± 1.06	7.67 ± 1.67	<0.001
HGI	−0.50 ± 0.38	0.43 ± 0.32	−0.81 ± 0.86	0.92 ± 0.87	<0.001

Data are presented as M ± SD or N(%), as appropriate. Statistical significance was defined as *p* < 0.05 under two-tailed condition. Abbreviations: CVD, cardiovascular disease; eGFR, estimated glomerular filtration rate; FPG, fast plasma glucose; TC, plasma total cholesterol; TG, triglyceride; HDL-C, high-density lipoprotein cholesterol; LDL-C, low-density lipoprotein cholesterol; SBP, systolic blood pressure; DBP, diastolic blood pressure; BMI, body mass index; HbA1c, glycated hemoglobin; HGI, hemoglobin glycation index; SD, standard deviation; M, mean.

**Table 2 jcm-11-05814-t002:** The association of HGI and the risk of incident stroke.

Variable	Incidence of Stroke			
	Crude HRs		Adjusted HRs	
	HR (95% CI)	*p*	HR (95% CI)	*p*
Continuous				
per SD of HGI	1.284 (1.047–1.574)	0.022	1.279 (1.063–1.539)	0.009
Categorical				
HGI < 0	reference	-	reference	-
HGI ≥ 0	1.977 (1.221–3.205)	0.020	1.780 (1.120–2.830)	0.015

Cox proportional hazards models include age, gender, current smoking and drinking, hypertension, DM, eGFR, TC, TG, HDL-C, LDL-C, and HGI. Statistical significance was defined as *p* < 0.05 under two-tailed condition. Abbreviations: HR, hazard ratio; SD, standard deviation; CI, confidence interval; HGI, glycated hemoglobin index.

**Table 3 jcm-11-05814-t003:** The value of HGI–DM system in predicting and evaluating the risk of stroke.

DM	High HGI	Events/Participants	Incidence of Stroke			
			Crude HRs		Adjusted HRs	
			HR (95% CI)	*p*	HR (95% CI)	*p*
−	−	21/1276	Reference	-	Reference	-
+	−	3/131	1.601 (0.478–5.368)	0.446	1.138 (0.337–3.847)	0.835
−	+	47/1344	1.966 (1.173–3.295)	0.039	1.701 (0.337–3.847)	0.036
+	+	8/183	2.566 (1.128–5.836)	0.025	2.655 (1.251–5.636)	0.011

Cox proportional hazards models include age, gender, current smoking and drinking, hypertension, eGFR, TC, TG, HDL-C, and LDL-C. The subjects were divided into 4 subgroups by HGI level and DM status. ‘+’ means positive and ‘−’ means negative. Statistical significance was defined as *p* < 0.05 under two-tailed condition. Abbreviations: HR, hazard ratio; SD, standard deviation; CI, confidence interval; HGI, glycated hemoglobin index; DM, diabetes mellitus.

**Table 4 jcm-11-05814-t004:** Reclassification analysis of HGI in improving conventional stroke model.

Incident Stroke	IDI	NRI
Conventional model	-	-
Conventional model+ HGI	0.012 (0.007–0.015)	0.036 (0.0198–0.0522)
*p* value	<0.001	<0.001

Conventional model includes age, gender, current smoking and drinking, hypertension, DM, eGFR, TC, TG, HDL-C, and LDL-C. Abbreviations: IDI, integrated discrimination improvement; NRI, net reclassification index; HGI, glycated hemoglobin index.

## Data Availability

We can provide the raw data of the present study after evaluation and permission from the subject principals. For matters on the availability of raw data, please contact Professor Xingang Zhang (zhangxingang80@aliyun.com) and Professor Yingxian Sun (yxsun@cmu.edu.cn).

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
