# Peer review of "The Value of Hemoglobin Glycation Index–Diabetes Mellitus System in Evaluating and Predicting Incident Stroke in the Chinese Population"

_jcm, 2022, doi:10.3390/jcm11195814_

Round 1

Reviewer 1 Report

This is an interesting study following up a rural population for a considerable time combining HGI and DM to evaluate the glucose status for predicting incident stroke. The manuscript needs considerable revisions for grammar, spelling checks and formatting.

47:Define DM.

Some abbreviations have not been defined when mentioned for the first time.

Please recheck the grammar and spelling of the whole manuscript.

102: Figure title should be under the flow diagram.

102: According to the diagram, 79 devolved stroke/dead. Mention the time duration

What were the inclusion and exclusion criteria

120: paper vision informed consent form??? Did you mean the paper version???

Line 127: What do you mean by variables measurements? Rephrase the title (line 127)

Line 132: Calibrate all laboratory equipment and repeat samples using blind method. This sentence is not complete.

133-134: rephrase the sentence.

Change told to asked.

Some of the sentences are not complete, please recheck.

143: what is JNC7. Define it.

154: Adjudication of Endpoints ?? rephrase the sub-topic to be clearer

155:median follow-up period

166-168:setence has to be rephrased. Please check the grammar.

196:However, the difference of these parameters between HGI groups were slightly…part of the sentence is missing

 In the table, change age values to one decimal point.

In the table, the first column variable should be left aligned.

199: statistically significance

207: During 4.23 years of following-up…during 4.2 years of follow-up period…

218: Figure titles/legends should come below the figure.

234:DM did not have a significantly increased risk….

What are the implications of these results in clinical practice? Worth mentioning in the discussion.

Author Response

Thank you very much for your comments concerning our manuscript entitled “The value of hemoglobin glycation index-diabetes mellitus system in evaluating and predicting the incident stroke in Chinese population” (Manuscript ID: jcm-1899297). Those comments are all valuable and very helpful for revising and improving our paper. We’d like to express our heartfelt thanks to you for your guidance and we totally agree with your suggestion. With your guidance, we revised the relevant issues in the manuscript, which greatly improved the quality of the article. Once again, we thank you for giving us such meticulous help in this article. The detailed correction in the attached file.

Reviewer 2 Report

This is a prospective cohort study aimed to investigate the association between HGI index and risk of incident stroke. The findings showed that higher HGI levels increased the risk of stroke and concluded that HGI-DM system could improve the predict ability of incident stroke based on traditional model. This article is informative, and some minor suggestion need to be revised.

1.      Please explain why so many 6858 subjects were excluded from this study? It might lead to selection bias.

2.      With regard to outcomes, the article only mentions new stroke events during follow-up, whereas there is another outcome, stroke-related death, mentioned in Figure 1 and in the Discussion. Please unify the manuscript. It is important because the incidence rate of stroke was 276.7 per 100 000 population in 2019 in China (https://svn.bmj.com/content/early/2022/04/20/svn-2021-001374) which is lower than this study (crude incidence rate: 6.30 incident stroke events per 1000 person-year)

3.      Some paragraph seems redundant, for example, study population and ethics and ethics approval and informed consent in p2 and p3. Also, line 220-225 in p8 is redundant. Please check the whole manuscript.

Author Response

Thank you very much for your comments concerning our manuscript entitled “The value of hemoglobin glycation index-diabetes mellitus system in evaluating and predicting the incident stroke in Chinese population” (Manuscript ID: jcm-1899297). These comments were very helpful for revising and improving our paper, as well as the important guiding significance to our research. We have studied the comments carefully and made corresponding corrections, which we hope meet with your approval. The detailed correction in the attached file.

Reviewer 3 Report

This study conducted a cohort in rural regions of Northeastern China to study the effects of the HGI-DM system on stroke risk. They found that DM-high HGI condition significantly elevated the risk of incident stroke. Moreover, higher HGI levels also elevated the risk of stroke, even if the patients didn’t have DM. However, there are many concerns that need to be addressed, and their results are not convincing. Below are my comments:

1.       From Table 1, we can see that except for the smoking factor, the P-values of other factors are smaller than 0.05, indicating that these factors may also affect the evaluation of HGI-DM system. If the authors cannot clarify or eliminate the impact of these factors, the downstream analysis is not convincing. I think at least the authors should at least test if some of these factors also affect the incidence of onset stroke.

2.       [Line 227] It’s better to draw Kaplan-Meier curves to better compare the stroke events between the four subgroups of HGI-DM in the section “High HGI could elevate the risk of incident stroke and promote adverse outcomes”

3.       What’s the difference between “Crude model” and “All factors adjusted model”?

4.       [Line 166] “Overall, the data was normally distribution.” Is there any test to support their conclusions? The authors should provide their test results here.

5.       [Line 176] “statistical software packages R”    R is only a program or software, please list the detailed R packages used in this study.

6.       Abbreviations should also be defined in the abstract.

7.       [Line 143] “According to JNC7, we defined hypertension as following conditions”   Please define the full name of JNC7 and list the reference if necessary.

Author Response

Thank you very much for your comments concerning our manuscript entitled “The value of hemoglobin glycation index-diabetes mellitus system in evaluating and predicting the incident stroke in Chinese population” (Manuscript ID: jcm-1899297). Those comments are all valuable and very helpful for revising and improving our paper. We’d like to express our heartfelt thanks to you for your guidance and we totally agree with your suggestion. With your guidance, we revised the relevant issues in the manuscript, which greatly improved the quality of the article. Once again, we thank you for giving us such meticulous help in this article. The detailed corrections are in the attached file.

Reviewer 4 Report

The study by Wang et al examined the association of the hemoglobin glycation index (HGI) in diabetic patients with the risk to the stroke. In order to do this they enrolled 2934 subjects form Chinese population. Their study concluded that HGI was associated with the stroke. The HGI-DM system help to detect individuals at risk to stroke based on traditional model.  This is assists in the identification and stratification of susceptible population for the purpose of prevention and treatment. The study is good but I have the following comments 

-it is a population based study, I would suggest that the title should be ''..................... evaluating and predicting the incident stroke in Chinese population

-The language of the should be revised carefully.

-in the introduction, the authors should define or introduce the  hemoglobin glycation index.

- in line 49, mention the abbreviation FPG levels in full for the first time. 

-In line 55, the HbA1C, the C should be a lowercase letter. 

- the authors should mention and discussed the related previous paper '' Hemoglobin Glycation Index Is Associated With Cardiovascular Diseases in People With Impaired Glucose Metabolism" by Chang Ho Ahn et al The Journal of Clinical Endocrinology & Metabolism, Volume 102, Issue 8, 1 August 2017, Pages 2905–2913, https://doi.org/10.1210/jc.2017-00191.

-the figure and table should be cited in the discussion as well.

Author Response

Thank you very much for your comments concerning our manuscript entitled “The value of hemoglobin glycation index-diabetes mellitus system in evaluating and predicting the incident stroke in Chinese population” (Manuscript ID: jcm-1899297). Too many thanks for your guidance and we totally agree with your suggestion. Your advices have led us to realize what remains to improve on our experiment section. The detailed correction are in the attached file.

Round 2

Reviewer 3 Report

The authors have addressed all my concerns.